# Demodicosis Mite Detection in Eyes with Blepharitis and Meibomian Gland Dysfunction Based on Deep Learning Model

**DOI:** 10.3390/diagnostics15243204

**Published:** 2025-12-15

**Authors:** Elsa Lin-Chin Mai, Ya-Ling Tseng, Hao-Ting Lee, Wen-Hsuan Sun, Han-Hao Tsai, Ting-Ying Chien

**Affiliations:** 1Department of Ophthalmology, Far Eastern Memorial Hospital, New Taipei City 220, Taiwan; 2Department of Electric Engineering, Yuan Ze University, Taoyuan City 320, Taiwan; 3Department of Optometry, Yuanpei University of Medical Technology, Hsinchu 300, Taiwan; 4Department of Computer Science and Engineering, Yuan Ze University, Taoyuan City 320, Taiwan; 5Department of Ophthalmology, China Medical University Hospital, Taichung City 404, Taiwan; 6Graduate Program in Biomedical Informatics, Yuan Ze University, Taoyuan City 320, Taiwan; 7Innovation Center for Big Data and Digital Convergence, Yuan Ze University, Taoyuan City 320, Taiwan

**Keywords:** Demodex mite, demodicosis, blepharitis, MGD, deep learning, object detection, AI

## Abstract

**Background/Objectives**: Demodex mites are a common yet underdiagnosed cause of ocular surface diseases, including blepharitis and meibomian gland dysfunction (MGD). Traditional diagnosis via microscopic examination is labor-intensive and time-consuming. This study aimed to develop a deep learning-based system for the automated detection and quantification of Demodex mites from microscopic eyelash images. **Methods**: We collected 1610 microscopic images of eyelashes from patients clinically suspected to have ocular demodicosis. After quality screening, 665 images with visible Demodex features were annotated and processed. Two deep learning models, YOLOv11 and RT-DETR, were trained and evaluated using standard metrics. Grad-CAM visualization was applied to confirm model attention and feature localization. **Results**: Both YOLO and RT-DETR models were able to detect Demodex mites in our microscopic images. The YOLOv11 boxing model revealed an average precision of 0.9441, sensitivity of 0.9478, and F1-score of 0.9459 in our detection system, while the RT-DETR model showed an average precision of 0.7513, sensitivity of 0.9389, and F1-score of 0.8322. Moreover, Grad-CAM visualization confirmed the models’ focus on relevant mite features. Quantitative analysis enabled consistent mite counting across overlapping regions, with a confidence level of 0.4–0.8, confirming stable enumeration performance. **Conclusions**: The proposed artificial intelligence (AI)-based detection system demonstrates strong potential for assisting ophthalmologists in diagnosing ocular demodicosis efficiently and accurately, reducing reliance on manual microscopy and enabling faster clinical decision making.

## 1. Introduction

Demodex mites are intradermal parasites that multiply in the hair follicles and sebaceous glands of humans and animals. They mostly spread through direct skin to skin contact. A previous study showed that more than 80% of people over the age of 60 and nearly all people over the age of 70 are affected by Demodex in the United States, bringing the total number of infected people to 25 million [1,2]. A systematic review paper reported the prevalence of ocular Demodex to be between 29% and 90% [3].

Demodicosis, or Demodex-associated dermatosis, is a chronic inflammatory condition caused by an overpopulation of Demodex mites. Although Demodex folliculorum and Demodex brevis are part of the normal human skin microbiota, under certain conditions such as immune dysregulation, aging, or skin barrier dysfunction, they may proliferate excessively, triggering cutaneous and ocular disease manifestations [1,3]. As people get older, the number of Demodex mites around their eyes tends to rise. This increase has been linked to various ocular surface problems, including MGD, blepharitis, chalazion and dry eye symptoms [4]. In addition, systemic conditions such as type 2 diabetes mellitus (DM) have been shown to increase the risk and severity of MGD. A recent meta-analysis of 13,629 participants reported a significant association between DM and MGD [5], and a cross-sectional study further demonstrated that patients with both dry eye and type 2 diabetes exhibit more severe lid margin irregularity, gland orifice plugging, meibomian gland dropout, and tear film instability compared with patients with non-diabetic dry eye [6].

Ocular demodicosis is a condition in which Demodex mites inhabit the anterior structures of the eye, including the eyelids, eyelashes (Figure 1 and Appendix A provided in Appendix A), and ocular surfaces. Comorbid diseases include MGD, chronic blepharitis, ocular surface inflammation, and dry eye [3,7,8,9]. Untreated demodicosis can lead to permanent changes in the eyelid margin, madarosis (loss of lashes) or superficial corneal pathologies, dry eye disease, neovascularization of the cornea, and marginal ulcers leading to vision loss [10]. One article reports the association between blepharitis and the development of ocular inflammation after cataract surgery, but the authors did not specify if they are directly related to Demodex [11].

Research has shown that ocular demodicosis is one of the potential causes of MGD with further disturbance of the tear film and eventual dry eye [12]. As seen in our demodicosis patients, Demodex infection leads to a collar of tissue around the base of the eyelashes, observed in a slit-lamp examination, and this finding is known as collarettes (Figure 1a,b). The microscopic view of this clinical finding is composed of epithelial hyperplasia, lipid materials, and mite waste products expressed on the opening of the lash follicle (Figure 1c,d).

In recent years, the rapid development of deep learning technology has led to revolutionary changes in various fields, especially in medicine. A significant number of resources have been invested in medical image analysis based on deep learning to assist physicians in rapid examination and diagnosis [13,14,15]. It is widely applied in fields such as cellular biology, medical imaging, and materials science to improve the automation and accuracy of image analysis. For example, Razzak et al. explored the optimization of deep learning for image segmentation and classification [16]. Chan et al. applied deep learning-enhanced image analysis for assisted diagnosis [17]. Gulshan and his team developed new algorithms using deep learning to automatically detect diabetic retinopathy and diabetic macular edema in retinal photographs [18]. Mai et al. utilized the cataract shadow projection theory and validated it by developing a deep learning algorithm that enables automatic and stable posterior polar cataract (PPC) screening using fundus images. This can increase safety in cataract surgery [13]. Swiderska et al. devised a deep learning approach for the evaluation of MGD [19]. With improvement in the AI deep learning approach by utilizing Meibography images, Li et al. reviewed AI usage in MG evaluation from slit-lamp photos, infrared imaging, confocal microscopy, and optical coherence tomography images [20]. Yu YW et al. introduced an automated detector chip for age-related macular degeneration (AMD) using a support vector machine (SVM) and three-dimensional (3D) optical coherence tomography (OCT), enhancing the automated eye disease detection system beyond DM retinopathy detection [21].

As the ability of computers to process images has improved from low-resolution to super-resolution imaging, this transition has also been achieved in deep neural networks. Microscopy images have also been utilized in combination with deep learning models: Ge and colleagues summarized the applications of deep learning analysis in microscopic imaging [22]; Von Chamier and his team proposed a platform, ZeroCostDl4Mic, to enhance the processing and analysis of microscope images [23]; Koohbanani et al. used microscopy images of nuclei and cells for analysis [24]; Weiß et al. documented hydrate samples with microscope images to ensure their structural integrity [25]; and Meijering et al. utilized microscope images to analyze various neural networks and employed deep learning for image analysis [26].

## 2. Materials and Methods

### 2.1. Study Design and Data Acquisition and Preprocessing

This study was conducted in Far Eastern Memorial Hospital in collaboration with Yuan Ze University and included data processing, image annotation, feature extraction, training, and validation. Data collection was carried out at the ophthalmology clinic through a structured effort involving ophthalmology residents extracting patient’s eyelashes after lash manipulation and photographing them under a microscope; the microscopic pictures were then uploaded to a database-labeled file. The image data used in this study consisted of microscopic images of the base of patients’ eyelashes, inhabited by the eyelash mite (Demodex) with four pairs of legs and an elongated tail, as shown in Figure 2. These images, captured using a high-magnification microscope (MICROTECH LX130.WF microscope, M&T OPTICS, Taipei, Taiwan), clearly revealed the fine structure of the base of the eyelash and the morphological characteristics of the Demodex mite, such as its elongated body and the distribution of its legs. These images were then cleaned and normalized by eliminating images that were out of focus, and good-quality images were processed for the recognition of Demodex mites via deep learning methodology.

### 2.2. Data Processing, Annotation, and Cropping

We gathered a raw dataset of 3422 images from the hospital clinic microscopic photography data for retrospective chart review, and these images were initially categorized into three groups: images in which Demodex mites could be identified, images showing only the eyelashes as a background, and images in which the mite features were unclear or blurred due to inaccurate focus and were deleted. A total of 1610 images of eye lashes with or without mites were obtained. These images were microscopic images of the eyelash base from various angles. Out of the acquired 1610 images, a total of 665 eyelash images containing clearly identifiable Demodex mites were retained for model development. Among these, 596 images contained a single mite, 51 images contained two mites, and 18 images contained three mites, indicating that most samples presented only one mite per image, while multi-mite occurrences were comparatively rare. LabelMe was used to mark the mites’ features in the images with polygonal labels to ensure that the marking range can completely cover the key parts of the mites. The four outermost points in the polygonal labels were selected and converted into square labels to ensure that the marking range is regular and convenient for subsequent processing. After determining the position of the Demodex, the four sides of the label were extended outward to 640 × 640 as the size of the cropped image. This size was the best image size for model training. If the Demodex label in the original image was larger than 640 × 640, the images were cropped in a square proportion and then scaled to 640 × 640 to ensure that the images contained the complete Demodex while ensuring a uniform image size. The data processing flowchart is given in Figure 3.

### 2.3. Dataset Augmentation, Division, and Confirmation

After preprocessing the image data, a total of 665 images were retained and subsequently divided into training, validation, and test datasets. However, due to the lack of metadata containing patient identifiers associated with each microscopy image, strict patient-level separation could not be implemented. Consequently, the datasets were assigned using randomized splitting. To ensure optimal results, the datasets were divided as follows: 15% of the total dataset (101 images) were in the test dataset; 70% of the total dataset (464 images) were in the training dataset; and 15% of the total dataset (100 images) were in the validation dataset, as shown in Figure 4.

To reduce the possibility of model distortion due to image size variations, all images and labels were resized to a uniform square size of 640 × 640. Due to the limited number of original microscopy images, we applied data augmentation to improve model robustness and reduce overfitting. Augmentation has been widely recognized as an essential strategy in medical imaging deep learning, particularly in scenarios where data collection is constrained by clinical workflow or privacy limitations. As emphasized in several recent reviews, augmentation enhances the diversity of training samples and improves generalization performance across imaging modalities and clinical tasks [27,28,29]. In this study, we augmented the training dataset with rotation, mirroring, and mirror flipping, increasing the number of training images to 3248, as shown in Figure 5.

In the training and validation stages, our primary objective was to teach the model to recognize the visual appearance and morphological patterns of Demodex mites. Therefore, both the training and validation sets consisted exclusively of images containing visible Demodex. However, in real-world clinical settings, eyelash microscopy images may frequently contain only eyelashes without any mites. To ensure that the model could correctly identify such negative cases and avoid misclassifying other eyelash structures as mites, we additionally included 103 ‘eyelash-only’ background images in the test set. This allowed us to evaluate whether the YOLOv11 and RT-DETR models could reliably distinguish true Demodex-positive images from Demodex-negative images. In these rounds of testing, the models did not detect mites in photos with only eyelashes but did in those with mites, and the models annotated a mite with a box and reported the odds of the image, such as an odd of 0.75, and no incorrect boxing was observed as the models correctly judged the absence of mites in an image.

### 2.4. Training Results

This study initially trained the models for 300 epochs. With the same number of training epochs, YOLOv11’s boxing detection method achieved good convergence results (Figure 6). YOLOv11’s segmentation detection method still had room for improvement in convergence at 300 epochs, but after adding an additional 100 epochs, it achieved even better convergence results (Figure 7). The RT-DETR model, initially trained for 300 epochs, achieved early stopping at 150 epochs (Figure 8).

### 2.5. Evaluation

This study used standard classification metrics (accuracy, sensitivity, and F1-score) to evaluate model performance. We defined images with Demodex features as positive data and further determined metrics such as accuracy, sensitivity, and F1-score (Formulas (1)–(3)).(1)$$Precision=\frac{TP}{TP+FP}$$(2)$$Sensitivity=\frac{TP}{TP+FN}$$(3)$$\text{F1-score}=\frac{2TP}{2TP+FP+FN}$$

Grad-CAM (version 1.5.4) was used to visualize the regions contributing to the model’s predictions. In this study, Grad-CAM was applied to the C3K2 layer of YOLOv11 and the basic block layer of RT-DETR. This approach was chosen to highlight the spatial attention of feature extraction layers rather than the detector head outputs. All our images are original and not generated by any generative artificial intelligence (GenAI) app. GenAI/ChatGPT (4.0) was used for text editing and writing augmentation only.

## 3. Results

In this study, the training, validation, and test sets were randomly assigned due to the characteristics of the available dataset. To ensure that the reported performance was stable and not dependent on a single random split, each model was trained and evaluated five independent times using different random seeds. This approach allowed us to assess the consistency of model behavior and verify the robustness of its predictions across multiple training–testing cycles. The following sections present the analysis results of each model in detail.

### 3.1. YOLOv11 Boxing

After multiple rounds of training, we obtained and analyzed the results of the two models. We present the results of the two models separately using tabular data. Figure 9a–c illustrate the results of each model’s testing.

We present three tables to evaluate our model’s ability to predict the presence of Demodex across five independent training–testing runs. As the datasets were randomly assigned in each run, the overall performance was calculated by averaging the results across the five trials. Table 1 shows the average and standard deviation of the accuracy, sensitivity, and F1-score of YOLOv11’s boxing detection method. Due to the presence of local occlusion or blurred boundaries in the image data, which can cause detection errors, the FP and FN values are slightly higher, but the overall data still demonstrate the stability of the segmentation detection method. The remaining data directly demonstrates the performance of the boxing detection method on various metrics.

### 3.2. YOLOv11 Segmentation

Table 2 shows the average and standard deviation of the accuracy, sensitivity, and F1-score of YOLOv11’s segmentation detection method. The model’s prediction evaluation results are as follows: a precision of 0.9331, a sensitivity of 0.9400, and a F1-score of 0.9363. Due to the presence of local occlusion or blurred boundaries in the image data, which can cause detection errors, the FP and FN values are slightly higher, but the overall data still demonstrate the stability of the segmentation detection method. These data demonstrate that YOLOv11 segmentation detection performs well in both precision and sensitivity, with overall stable performance, effectively improving the integrity of Demodex detection.

### 3.3. RT-DETR

Due to noise and unclear boundaries in the data, the model has some errors in determining FPs and FNs, but the overall performance demonstrates good detection capabilities. Table 3 shows the model prediction evaluation results. The average precision, sensitivity, and F1-score were 0.7513, 0.9389, and 0.8322, respectively. Overall, RT-DETR performed well in terms of sensitivity, and the F1-score value shows its stable detection performance.

### 3.4. Grad-CAM Feature Evaluation

We used Gradient-weighted Class Activation Mapping (Grad-CAM) to evaluate how the model detected Demodex and to determine characteristics that were the weighted distributes. Analysis of the heatmaps revealed that both the boxing and segmentation methods in the YOLOv11 model correctly detected Demodex characteristics. The RT-DETR model also accurately detected Demodex and its characteristics. Although some background noise affected the model’s judgment, overall, the model performed at a reasonable level. Figure 10a–c illustrate the Grad-CAM results for the boxing and segmentation methods of the YOLOv11 model and the RT-DETR model.

### 3.5. Demodex Quantitative Analysis

Our model demonstrated stable detection capabilities in areas with overlapping Demodex. Figure 11 shows the number of Demodex predicted by the YOLOv11 boxing and segmentation detection methods, as well as the RT-DETR model, with a confidence level of 0.42~0.88. YOLOv11’s boxing detection method successfully detected two Demodex points in Figure 11a, while the segmentation detection method also accurately detected Demodex points in Figure 11b. The RT-DETR model detected three Demodex points in Figure 11c. Overall, YOLOv11 excels in boundary detection, while RT-DETR maintains good consistency in complex backgrounds and overlapping conditions, demonstrating the practical value of both methods in quantitative analysis.

## 4. Discussion

In this study, we leveraged the YOLOv11 and RT-DETR models to discern the presence of Demodex at the base of eyelashes. Our model achieved an average precision of 0.9442, sensitivity of 0.9478, and F1-score of 0.9459 in our detection system. These results, though variable, indicate a substantial capacity to correctly identify Demodex and highlight potential for further refinement. Additionally, by pinpointing the exact location of mite infestation, the AI model not only aids in diagnosis but also enhances the clinical management of related eye diseases. During this study, we also experimented with YOLOv8 [30] and YOLOv9 [31]. YOLOv8 introduced anchor-free detection to improve small-object sensitivity; YOLOv9 introduced programmable gradient information (PGI) and the generalized efficient layer aggregation network (GELAN) [31]. However, due to the limited number of images, the model suffered from overfitting, and we were unable to achieve conclusive results. Future studies with larger datasets may enable better comparison between the models.

Several methods are available for the clinical diagnosis of ocular Demodex. The most popular method is the microscopic examination of epilated eyelashes and their subjective counting for quantification. Other methods include a direct slit-lamp examination after the removal of collarettes, in vivo confocal microscopy (IVCM) lash follicle scanning, and procedures such as applying lateral tension to eyelashes without epilation, and rotating lashes clockwise and counterclockwise with forceps under slit-lamp observation [3]. Currently, in our hospital, demodicosis is diagnosed based on a high index of clinical suspicion. Ophthalmologists usually examine the eye under a slit lamp, looking for signs of meibomian gland dysfunction (MGD) and blepharitis such as lid margin redness, foamy tears, lid margin notching and irregularity. More specific signs of demodicosis include thickened or capped meibomian orifices, collarettes or cylindrical dandruff at the lash base, and eyelashes that easily fall out. For definite proof of mites, lashes are manipulated before removal and examined under a microscope (16×–40×). Clinicians can also check for moving mites near the follicle of the lash. Although these processes are time-consuming and tedious, it remains necessary for the definite diagnosis of demodicosis.

Detection of Demodex mites has been a key focus in clinical practice for blepharitis, given their established role as a primary contributor to chronic and refractory cases [3]. However, diagnosis varies significantly across countries, partly due to reliance on examiner expertise and the time-intensive nature of current diagnostic techniques. For example, in Nikhil Sharma’s study, the estimated perceived prevalence of Demodex infestation was reported as 60% in Australasia compared to 27% in India (*p* < 0.01). While nearly 70% of practitioners recognized Demodex-associated blepharitis, only 45% reported actively attempting to identify Demodex in their patients [32]. Despite this, the microscopic examination of epilated lashes remains the most widely used, cost-effective method [33]. Other advanced techniques, such as IVCM, enable the real-time visualization of mites in sebaceous and meibomian glands, but their utility is limited by a high level of required expertise and being highly time-consuming [34]. Molecular techniques like PCR and digital PCR offer higher sensitivity (93.75% for face and 86.7% for scalp) [35,36], but their high cost and lack of quantification hinder routine clinical use. Novel methods, including smartphone-aided intraocular lens tools, have also been explored for rapid clinic-based detection [37].

To address these diagnostic hurdles, we devised this study to augment our clinical workflow. Patients with suspected anterior blepharitis due to mite infection underwent eyelash manipulation and the selective epilation of lashes with collarettes. These lashes were examined under a microscope, and the resulting images were stored in a database for annotation, convolutional training, and validation. Our AI-assisted detection method demonstrated encouraging results, showing that convolutional neural networks (CNNs) can provide both qualitative and quantitative detection outputs.

Basic lid hygiene with warm compresses and tea tree oil scrubs remains the first-line therapy for demodicosis. In more severe cases, in-clinic procedures such as BlephEx^®^ manipulation or intense pulsed light (IPL) are performed. Patients are instructed on self-care and maintenance therapy, follow-up visits with repeat lash epilation and mite quantification are required to assess improvement or eradication. Accurate mite quantification facilitated by AI could provide a standardized and objective measure to evaluate treatment efficacy, thereby supporting clinicians in tailoring therapy.

Despite these advances, limitations exist. Due to a lack of manpower and resources, not all patients with blepharitis can undergo epilation and microscopic examination. Our dataset was limited to 3422 images, with approximately half of the images discarded due to poor contrast, which likely impacted sensitivity and accuracy. This reflects a common challenge in AI model development, where data quality and quantity are critical factors.

Future research should focus on developing better and faster CNN models for qualitative and quantitative Demodex detection, integrating segmentation approaches to label entire mites rather than only distinctive features. With more robust datasets, AI systems could provide more accurate quantification, enabling clinicians to evaluate treatment efficacy—such as tea tree oil scrubs, lid hygiene, or IPL—more precisely. Additionally, exploring non-invasive techniques and multimodal AI-assisted tools may further streamline workflows and improve early detection of demodicosis.

Overall, our AI-assisted detection method shows promise in addressing the time-intensive nature of conventional diagnosis, providing standardized and objective outputs, this AI application can be utilized by automate reading of microscopic slides with eye lash specimens taken by technicians and instantaneously providing a quantitative and qualitative answer to the clinical staffs, therefore ultimately reducing the workload on ophthalmology staff while improving patient management and educations.

## 5. Conclusions

In busy ophthalmology clinics, definite demodicosis diagnosis relies on a direct lash examination under a microscope in a laboratory setup. This labor-intensive procedure takes up precious time and resources from residents and young ophthalmologists. Our automated AI model provides both qualitative and quantitative analyses, reducing diagnostic workload and enabling the objective assessment of Demodex infestations. AI can also help in determining the quantitative amounts of infestations for patient education and treatment improvements.

## Figures and Tables

**Figure 1 diagnostics-15-03204-f001:**
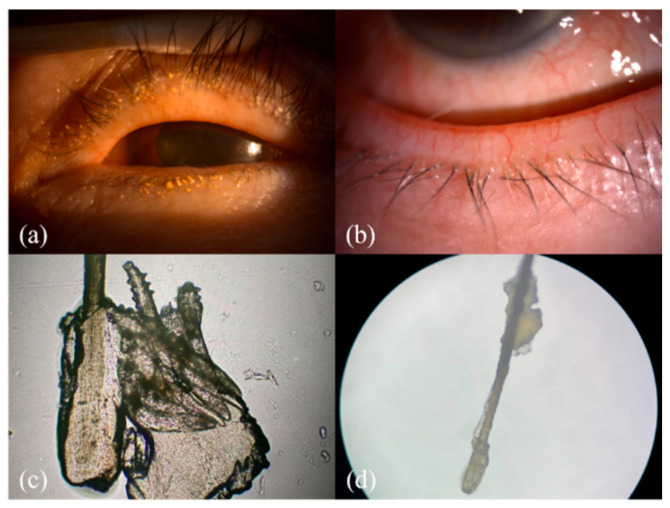
The clinical and microscopic findings of demodicosis: (**a**) multiple Demodex mite infestations on the follicle of the eye lash view by slit lamp; (**b**) collarettes of Demodex mite infestation observed with a slit-lamp examination on the lower lid margin; (**c**,**d**) collarettes on the lash composed of epithelial hyperplasia and lipid materials along with the infestation of Demodex mites (100× & 40× microscopic view of the epilated lash).

**Figure 2 diagnostics-15-03204-f002:**
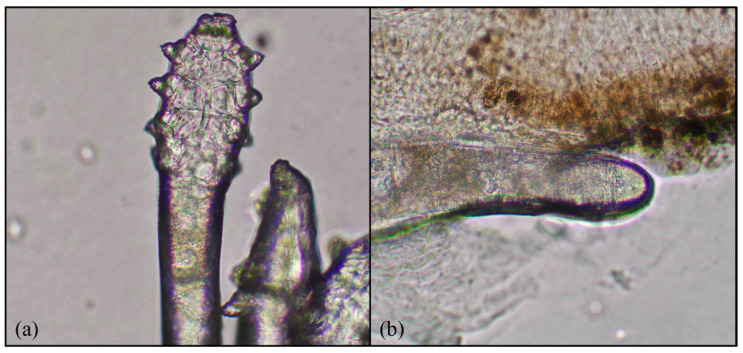
Demodex features: (**a**) four pairs of legs and (**b**) an elongated body. Magnification 64×.

**Figure 3 diagnostics-15-03204-f003:**
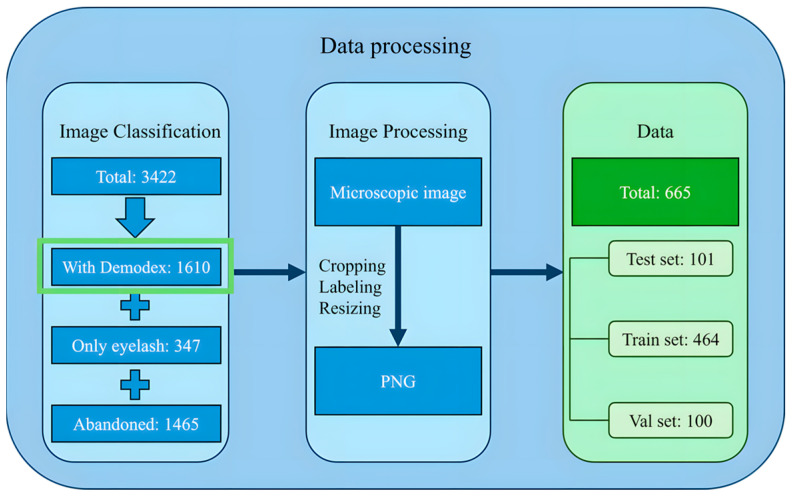
The data processing workflow.

**Figure 4 diagnostics-15-03204-f004:**
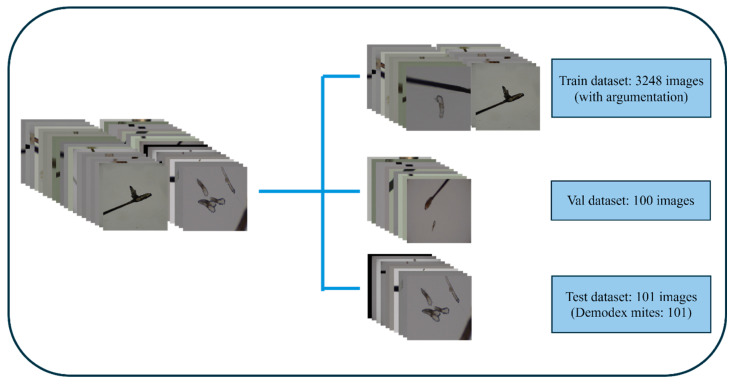
A schematic diagram of dataset allocation into 3 groups. Val: validation.

**Figure 5 diagnostics-15-03204-f005:**
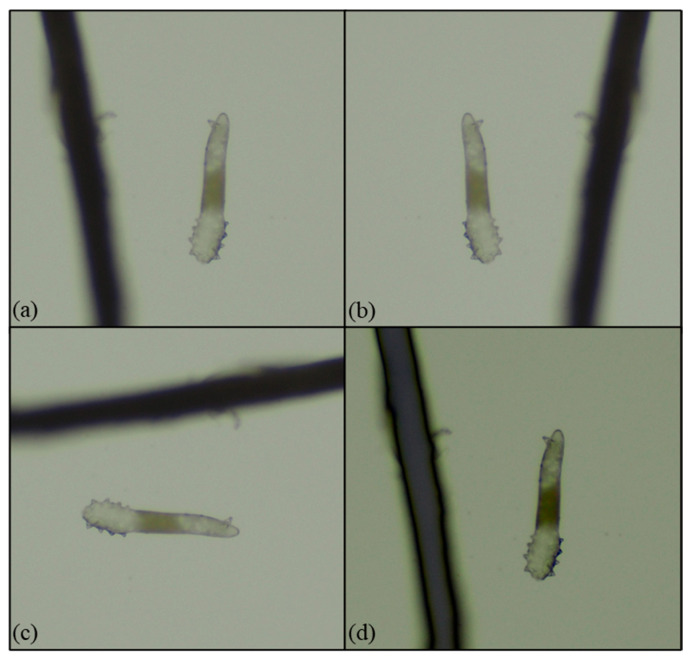
Data augmentation: (**a**) a microscopy image (40×); (**b**) a mirror microscopy image; (**c**) a 90-degree right-rotated microscope image; and (**d**) a dimmed microscopy image.

**Figure 6 diagnostics-15-03204-f006:**
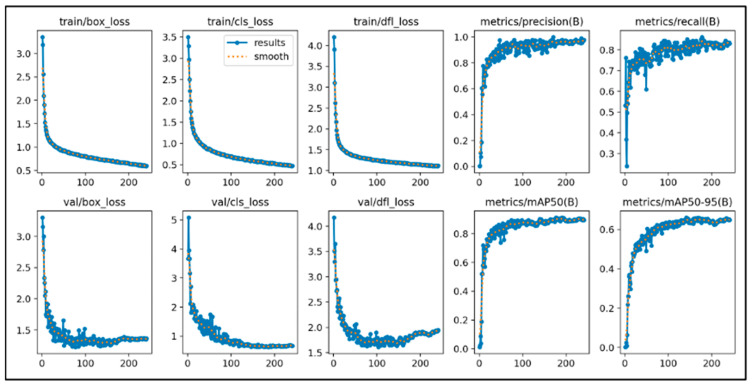
The learning curve of Yolov11’s boxing.

**Figure 7 diagnostics-15-03204-f007:**
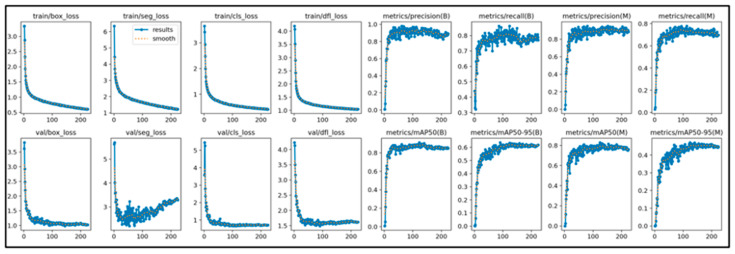
The learning curve of Yolov11’s segmentation.

**Figure 8 diagnostics-15-03204-f008:**
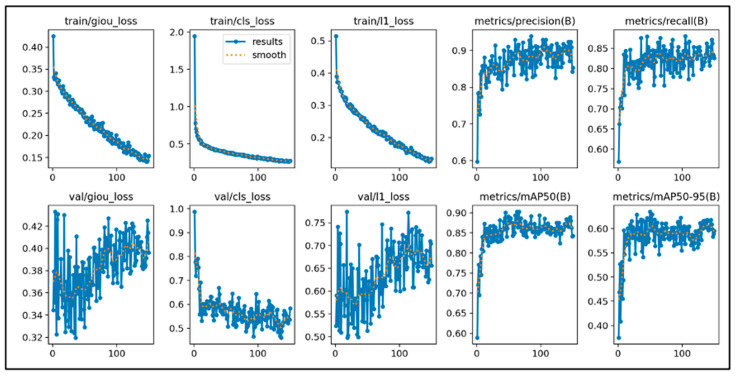
The learning curve of RT-DETR.

**Figure 9 diagnostics-15-03204-f009:**
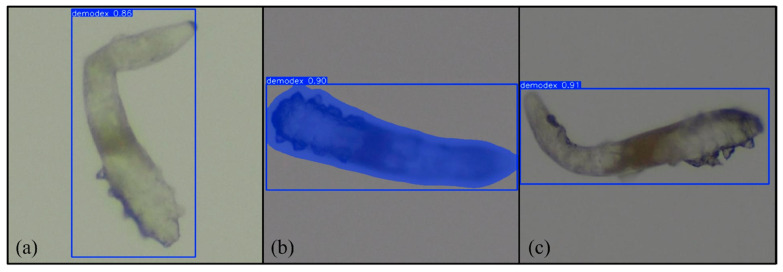
Results of (**a**) YOLOv11 boxing, (**b**) YOLOv11 segmentation, and (**c**) RT-DETR. Magnification 64×.

**Figure 10 diagnostics-15-03204-f010:**
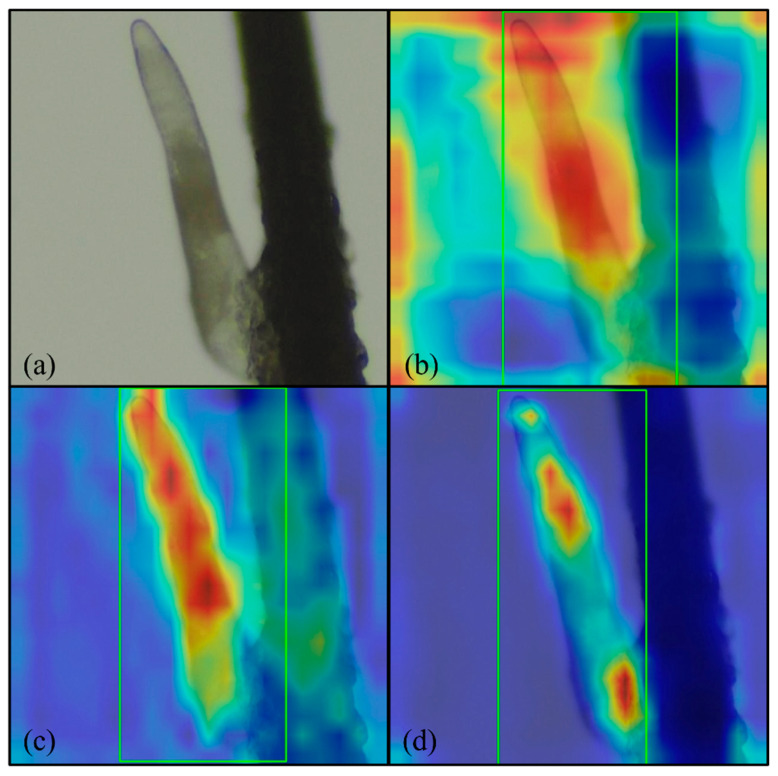
Grad-CAM results: (**a**) original image; (**b**) YOLOv11 boxing; (**c**) YOLOv11 segmentation; and (**d**) RT-DETR. Magnification 64×, green rectangle: Demodex detected by proposed model.

**Figure 11 diagnostics-15-03204-f011:**
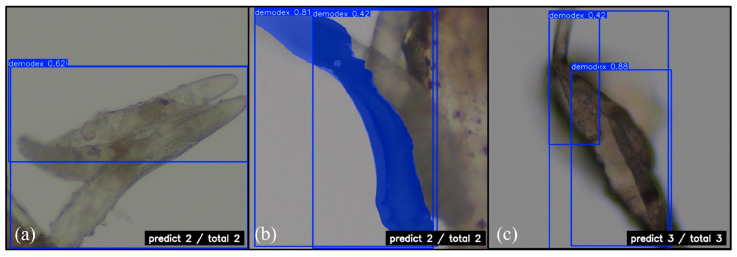
Quantitative prediction of Demodex mite: (**a**) YOLOv11 boxing; (**b**) YOLOv11 segmentation; and (**c**) RT-DETR. Magnification 64×.

**Table 1 diagnostics-15-03204-t001:** Averaged prediction performance across five independent runs for the YOLOv11 boxing model.

	Average	Standard Deviation
Precision	0.9442	0.020
Sensitivity	0.9478	0.019
F1-score	0.9459	0.017

**Table 2 diagnostics-15-03204-t002:** Averaged prediction performance across five independent runs for the YOLOv11 segmentation model.

	Average	Standard Deviation
Precision	0.9331	0.022
Sensitivity	0.9400	0.012
F1-score	0.9363	0.010

**Table 3 diagnostics-15-03204-t003:** Average prediction performance across five independent runs for the RT-DETR model.

	Average	Standard Deviation
Precision	0.7513	0.0418
Sensitivity	0.9389	0.0499
F1-score	0.8322	0.0126

## Data Availability

The raw data presented in this study are available on request from the corresponding author due to the sensitive medical chart issues and restriction of FEMH IRB.

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
