# Peer review of "Demodicosis Mite Detection in Eyes with Blepharitis and Meibomian Gland Dysfunction Based on Deep Learning Model"

_diagnostics, 2025, doi:10.3390/diagnostics15243204_

Round 1
Reviewer 1 Report
Comments and Suggestions for Authors
Current title implies general MGD diagnosis, but all data are microscopic images of epilated eyelashes and the task is Demodex detection/counting. Please revise to reflect: target organism, modality, and task.
Throughout the manuscript, MGD is discussed largely as an isolated eyelid disorder. However, systemic comorbidities—especially DM—materially influence meibomian gland structure and function and can confound both prevalence and algorithmic performance. A recent meta-analysis demonstrated a significant association between DM and MGD and recommends attention to imaging domain when interpreting effect sizes; please cite and integrate this evidence (The association between diabetes mellitus and meibomian gland dysfunction: A meta-analysis. Int J Diabetes Dev Ctries. 2025)
MGD is multifactorial; this work addresses one etiologic factor (Demodex) via a single imaging domain (microscopy).
Please avoid framing this as an “MGD detection” study. MGD involves gland obstruction/dysfunction modulated by age, hormones, environment, ocular surface inflammation, rosacea, iatrogenic factors, etc.; Demodex is one contributor. Where the text states “potential causes of MGD,” ensure you distinguish association from diagnosis and keep claims strictly to mite detection on microscopy.
The authors began with 3,422 raw images, retained 1,610 microscopy images, and finally used 665 images with visible Demodex for model development, with 70/15/15% splits and heavy augmentation. Please state:
a) whether patient-level splitting was enforced (to avoid the same patient contributing to train/val/test via multiple crops/angles); b) the exact composition of negative images in test/val (and whether any “eyelash-only background” images were included there); c) the distribution of mites per image (for counting difficulty).
Grad-CAM is more common for classifiers than for detector heads. Please specify which layer/branch the authors visualize (backbone vs. head).
Reviewer 2 Report
Comments and Suggestions for Authors
This manuscript presents an interesting and timely study combining ophthalmology and artificial intelligence for the automated detection of Demodex mites in microscopic eyelash images. The topic is relevant to both clinicians and AI researchers, as demodicosis remains underdiagnosed and current methods are labor-intensive. The study demonstrates practical potential for improving diagnostic workflows in ophthalmology.
However, while the concept is solid, the manuscript would benefit from significant improvements in scientific depth, contextualization, and reference support. The paper currently feels more like a technical report than a fully developed research article suitable for publication in Diagnostics.
Strengths
-
Novelty: The use of YOLOv11 and RT-DETR models for Demodex detection is innovative and original.
-
Clinical relevance: The integration of AI into ocular surface disease diagnosis has high clinical utility.
-
Clarity of presentation: The methods and figures are generally clear, and the Grad-CAM visualizations are helpful.
Weaknesses
a. Insufficient references
-
The article includes only 24 references, which is too few for an MDPI Diagnostics article.
-
Many statements (particularly in the Introduction and Discussion) are not supported by appropriate or recent citations.
-
References on Demodex biology, ocular surface inflammation, and AI-based diagnostic imaging are minimal.
-
To reach an acceptable academic standard, at least 35-40 references should be included, with emphasis on:
-
Recent AI applications in ophthalmology (2020–2025)
-
Machine learning for parasitic or dermatologic detection
-
Updated epidemiological data on ocular demodicosis
-
Modern imaging or diagnostic technologies (IVCM, PCR, etc.)
-
b. Limited discussion
-
The Discussion section lacks depth and critical comparison with previous AI or diagnostic methods.
-
The authors should explicitly compare their results with other object detection networks (YOLOv5, YOLOv8, Faster R-CNN, etc.) in medical contexts.
-
The potential biases introduced by small dataset size and manual annotation are mentioned only briefly and should be expanded.
c. Methodological details
-
The training dataset (665 images) is relatively small for deep learning. Data augmentation is mentioned but not sufficiently justified or evaluated.
-
No external validation or cross-institutional testing is performed, which limits generalizability.
-
The use of “YOLOv11” is confusing—this version is not officially released or recognized in the literature. Clarification is required.
The English is understandable but needs grammatical and stylistic editing for conciseness and clarity.
Round 2
Reviewer 1 Report
Comments and Suggestions for Authors
The authors well addressed all my concerns. Please briefly describe how it can be used more effectively in actual clinical settings.
Reviewer 2 Report
Comments and Suggestions for Authors
I have reviewed the revised version of the manuscript. The authors have addressed all the previous comments, and the necessary modifications have been made appropriately.
I recommend acceptance of the manuscript in its current form.
